# FLVCR1-AS1 and FBXL19-AS1: Two Putative lncRNA Candidates in Multiple Human Cancers

**DOI:** 10.3390/ncrna9010001

**Published:** 2022-12-22

**Authors:** Mohsen Sheykhhasan, Hamid Tanzadehpanah, Amirhossein Ahmadieh Yazdi, Hanie Mahaki, Reihaneh Seyedebrahimi, Mohammad Akbari, Hamed Manoochehri, Naser Kalhor, Paola Dama

**Affiliations:** 1Research Center for Molecular Medicine, Hamadan University of Medical Sciences, Hamadan 6517838636, Iran; 2Department of Mesenchymal Stem Cells, Academic Center for Education, Culture and Research, Qom 3716986466, Iran; 3Antimicrobial Resistance Research Center, Mashhad University of Medical Sciences, Mashhad 9177899191, Iran; 4Vascular & Endovascular Surgery Research Center, Mashhad University of Medical Sciences, Mashhad 9177899191, Iran; 5Anatomy Department, Faculty of Medicine, Qom University of Medical Sciences, Qom 3715614566, Iran; 6General Physician, Department of Medical School, Faculty of Medical Sciences, Islamic Azad University, Tonekabon Branch, Mazandaran 4684161167, Iran; 7School of Life Sciences, University of Sussex, Falmer, Brighton BN1 9QG, UK

**Keywords:** LncRNA, FLVCR1-AS1, FBXL19-AS1, cancer, molecular mechanisms, signaling pathways

## Abstract

(1) Background: Mounting evidence supports the idea that one of the most critical agents in controlling gene expression could be long non-coding RNAs (lncRNAs). Upregulation of lncRNA is observed in the different processes related to pathologies, such as tumor occurrence and development. Among the crescent number of lncRNAs discovered, FLVCR1-AS1 and FBXL19-AS1 have been identified as oncogenes in many cancer progression and prognosis types, including cholangiocarcinoma, gastric cancer, glioma and glioblastoma, hepatocellular carcinoma, lung cancer, ovarian cancer, breast cancer, colorectal cancer, and osteosarcoma. Therefore, abnormal FBXL19-AS1 and FLVCR1-AS1 expression affect a variety of cellular activities, including metastasis, aggressiveness, and proliferation; (2) Methods: This study was searched via PubMed and Google Scholar databases until May 2022; (3) Results: FLVCR1-AS1 and FBXL19-AS1 participate in tumorigenesis and have an active role in impacting several signaling pathways that regulate cell proliferation, migration, invasion, metastasis, and EMT; (4) Conclusions: Our review focuses on the possible molecular mechanisms in a variety of cancers regulated by FLVCR1-AS1 and FBXL19-AS1. It is not surprising that there has been significant interest in the possibility that these lncRNAs might be used as biomarkers for diagnosis or as a target to improve a broader range of cancers in the future.

## 1. Introduction

Cancer is among the world’s most deadly pathological diseases. In 2022, more than 1.9 million new cancer cases and 600 million cancer deaths were projected to occur in the United States [1]. Genetic and epigenetics agents play a key role in cancer incidence. Among these agents, non-coding RNAs (ncRNAs) in the last years have been described for their critical role in tumorigenesis and cancer progression [2,3]. Accordingly, to their functional actions, ncRNAs are distinguished in housekeeping RNAs, abundantly and ubiquitously expressed in cells to regulate generic cellular functions and regulatory RNAs involved at epigenetic, transcriptional, and post-transcriptional levels. They are further classified based on the average size of nucleotides. Long ncRNAs (lncRNAs) are one of the major sub-grouped regulatory RNAs defined by RNA transcript lengths larger than 200 nucleotides [4,5,6,7]. Based on their location with respect to protein-coding genes, lncRNAs are distinguished into long intergenic ncRNAs (lincRNAs), long intronic ncRNAs, sense or anti-sense lncRNAs, and bidirectional lncRNA. LncRNA can be transcribed by polymerase II and fold into second structures useful for the interactions with DNA, RNA, and protein, playing an active role in cancer regulating gene transcription through multiple mechanisms: Refs. [4,8] through chromatin and epigenetic remodeling by interaction and recruitment of chromatin-modifying enzymes to the target genes locus; Refs. [9,10,11,12,13] through the interaction with other RNA-binding factors to form RNA-protein complexes (RNPs) and through the recruitment of transcriptional machinery proteins to adjacent target gene locus [9,10,11,12,13].

LncRNAs can operate as decoys by binding to microRNAs or transcription factors to sequester them away from their target locations and obstruct transcription and translation [9]. The role of lncRNA as competing endogenous RNAs (ceRNAs) in the development of cancer has been demonstrated [7,9,14]. Through their regulatory effects on DNA sequences in cis-acting and trans-acting lncRNAs, the lncRNAs can modulate many biological processes such as cell growth, proliferation, differentiation, invasion, progression, apoptosis, epithelial–mesenchymal transition (EMT), and tumorigenesis [2,3,4,8,9]. There is growing proof that lncRNAs interact with DNA in sequence-specific ways by forming triple helix (triplex) structures [15]. The transcriptional factors bound on a specific DNA sequence that take control of the gene expression frequently interact with LncRNA [16]. On the other hand, LncRNA co-transcriptionally form RNA-DNA hybrids such as R-loops recognized by chromatin modifiers or by transcription factors to activate or inhibit target gene transcription [17].

Finally, the identification and detailed description of the lncRNAs involved in the occurrence and development of numerous distinct cancers may be applied to cancer detection and treatment. LncRNA dysregulation is present in the onset and progression of various cancers: Both upregulation and downregulation can play a role in the formation and spread of cancer [18,19,20]. Since lncRNAs in both tissues and fluids are easily detectable, they can be considered a suitable diagnostic biomarker in cancer diagnosis [21,22,23,24,25,26,27].

In this study, we present and discuss the data on two crucial lncRNAs named lncRNA feline leukemia virus subgroup C cellular receptor 1 anti-sense RNA 1 (FLVCR1-AS1) and lncRNA F-Box and leucine-rich repeat protein 19 anti-sense RNA 1 (FBXL19-AS1) upregulated in the various cancers.

The FLVCR1-AS1 gene (ID number NONHSAG004247.3 NONCODE databases) is 688 nt in length and is located on human chromosomal region 1q32.3, chr1:212,856,603-212,858,138 (Figure 1).

LncRNA may contribute to disease progression by splicing regulation. FLVCR1-AS1 gene has six splice variants (http://grch37.ensembl.org/Homo_sapiens/Gene/Summary?g=ENSG00000198468;r=1:213025450-213031430, accessed on 30 October 2022). In addition, According to the lncLocator database (http://www.csbio.sjtu.edu.cn/bioinf/lncLocator2/, accessed on 26 January 2021), FLVCR1-AS1 is located in the nucleus and cytoplasm (Table 1). FLVCR1-AS1 has been described as a new tumor suppressor as its impaired expression can lead to the development and spread of various cancers such as cholangiocarcinoma (CCA) [28], hepatocellular carcinoma (HCC) [29,30], gastric cancer (GC) [31], colorectal cancer (CRC) [32,33], glioma and glioblastoma (GBM) [34,35], non-small cell lung cancer (NSCLC) [36,37], ovarian cancer (OC) [38], breast cancer (BC) [39], and osteosarcoma (OS) [40], and in October 2021, was reported in pancreatic cancer (PC) [41].

The FBXL19-AS1 (gene ID number NONHSAG019125.3 NONCODE databases) is only one transcript of 7385 nt in length and is located on the human chromosomal region 16.p11.2 chr16:30915910-30923295 (http://grch37.ensembl.org/Homo_sapiens/Gene/Summary?db=core;g=ENSG00000260852;r=16:30930640-30934590;t=ENST00000563777, accessed on 30 October 2022) (Figure 2). According to the lncLocator database (http://www.csbio.sjtu.edu.cn/bioinf/lncLocator2/, accessed on 26 January 2021), FBXL19-AS1 is located in the nucleus and cytoplasm (Table 2).

The oncogenic role of lncRNA has been described in several cancers, such as hepatocellular cancer and cholangiocarcinoma [42,43,44,45], gastric cancer [46,47], colorectal cancer [48], glioma [49], lung cancer [50,51,52], cervical cancer (CC) [53,54], breast cancer [55,56,57], osteosarcoma [58], nasopharyngeal carcinoma (NPC) [59], and acute myeloid leukemia (AML) [60].

According to the last updated (September 2021) information by the Lnc2Cancer version 3 database (http://www.bio-bigdata.com/lnc2cancer, accessed on 30 September 2021), mean expression of FLVCR1-AS1 and FBXL19-AS1 in cancer and normal tissue is shown in Figure 3 and Figure 4.

Herein, it is our purpose to provide a concise review of current evidence associated with the role of FLVCR1-AS1, and FBXL19-AS1 in different cancers, by firstly discussing the abnormal expression of these lncRNAs in patient samples, followed by the description of their molecular mechanism of action and overall impact on signaling pathways leading to cancer development and progression.

## 2. Search Strategy and Literature Selection

This study was searched via PubMed and Google Scholar databases until May 2022. The following keywords were used in combination for searches: (“long non-coding RNA” OR “lncRNA”), (“FLVCR1-AS1” OR “Feline Leukemia Virus Subgroup C cellular receptor 1 antisense RNA 1”), (“FBXL19-AS1” OR “F-Box and Leucine-Rich Repeat Protein 19 anti-sense RNA 1”), AND (“carcinoma” OR “cancer” OR “neoplasm”). The reference lists of included articles were also screened for potentially missing literature.

## 3. The Effect of FLVCR1-AS1 on Several Cancers

### 3.1. Cholangiocarcinoma

Cholangiocarcinoma is a bile duct gastrointestinal highly heterogeneous cancer at genomic, epigenetic, and molecular levels with a limited outcome to surgical treatment, radiotherapy, and chemotherapy. To date, CCA, in most cases, is diagnosed when it is advanced, often unresectable, and combined with an aggressive nature, and because of CCA, early diagnosis is difficult. It has progressed into a poor prognosis. Their incidence is globally increasing. Therefore, identifying molecular and new epigenetic biomarkers in CCA development is a crucial diagnostic tool and can help with the efficacy of the available treatments [26].

A study investigated the oncogenic effect of FLVCR1-AS1 in regulating human CCA cell proliferation, migration, and invasion. The authors first demonstrated that the expression level of FLVCR1-AS1 was significantly increased in CCA tissue compared with the adjacent normal tissues and in CCA cell lines. Bioinformatics analysis and reporter assay revealed that the lncRNA FLVCR1-AS1 acts as an oncogene sponging the miR-485-5p. As the authors pointed out, it would be interesting to investigate the broad molecular mechanism proposing it as a novel diagnostic marker for CCA [28].

### 3.2. Hepatocellular Carcinoma

Hepatocellular carcinoma is a relatively common type of liver tumor. Numerous lncRNAs have been described as involved in the development and progression of HCC [21,29]. A study demonstrated that FLVCR1-AS1 plays an important function in the growth and progression of HCC by sponging miR-513c [30]. The LncRNA was highly upregulated in HCC tissues and cell lines. The FLVCR1-AS1 expression level was positively correlated with tumor severity, and its knockdown remarkably inhibited HCC cell proliferation, migration, and invasion in vitro and in vivo [30].

### 3.3. Gastric Cancer

Although the prevalence of gastric cancer can be reduced by good nutritional habits and treatment of Helicobacter pylori, it is still one of the most common cancers with high mortality [61,62]. Several studies have reported different miRNA expression changes in gastric cells, and this aberrant expression can regulate cell proliferation and metastasis [63,64,65,66]. Moreover, ectopic expression of lncRNAs regulates the gene expression both in the nucleus and cytoplasm, leading to gastric cancer progression [66]. In our recent review, we documented Lnc DLX6-AS1 as a crucial agent for the growth and proliferation of GC [67]. Liu et al. reported that an upregulated expression of FLVCR1-AS1 in gastric cancer cells by sponging miR-155 could increase the proliferation and invasion of gastric cells, implying its role in tumorigenesis [31].

### 3.4. Colorectal Cancer

Colorectal cancer is the second-most common cause of cancer death in men and women [1]. Diagnostic biomarkers for the early detection of CRC are still an unmet need. A pilot study pointed to the role of circulating lncRNAs as potential biomarkers. A total of 13 candidates upregulated in the plasma of 18 CRC patients were investigated. Interestingly the sensitivity in an early stage of lncRNA 91H, PVT-1, and MEG3, compared to the combination of CEA and CA19-9, currently in use for detection, was increased [22]. To shed light on the possible role of FLVCR1-AS1, in an experimental study, Han et al. compared 26 samples of CRC tissues with matched adjacent non-tumor tissues. They also measured the relative expression on CRC cell lines. lncRNA FLVCR1-AS1 resulted in upregulated in CRC cells and tissues; therefore, phenotypic assays showed a positive correlation on increasing CRC cell viability, migration, and invasion through regulation of the Mir 381/RAP2A axis. The authors suggested targeting FLVCR1-AS1 as a novel approach for treating CRC [32]. In another study, LncFLVCR1-AS1 acts as a miR-493-3p sponge to modulate cancer cell proliferation, invasion, and migration in CRC [33].

### 3.5. Glioma and Glioblastoma

Glioma originates from glial cells and has been identified as one of the most common fatal intracranial tumors. Despite the progress and the benefit in clinical stratification of patients due to understanding the molecular biology and genetics of GC, the impact on high mortality has not been affected. Thus, it is crucial to identify novel functional molecular targets involved in glioma development [68]. LncRNA FLVCR1-AS1 expression was upregulated in glioma tissue and cell lines and modulated E2F2 expression by sponging miR-4731-5p. The research showed the role of the FLAVR1-AS1/miR-4731-5p/E2F2 axis as a diagnostic marker of poor outcomes and as a potential target for tumor therapy [34]. Furthermore, another study has indicated that FLVCR1-AS1 is involved in the development and progression of glioma cancer. In that study, FLVCR1-AS1 expression was significantly upregulated in GBM tissues and cell lines compared with adjacent normal brain samples and human astrocyte cells, respectively. The authors, by luciferase reporter assay and rescue experiments, demonstrated that FLVCR1-AS1 acts as ceRNA for miR-30b-3p, suppressing cellular proliferation and invasion in various types of cancer, including gliomas. In addition, this article’s conclusion led to the use of FLVCR1-AS1 as a novel therapeutic target and diagnostic biomarker [35].

### 3.6. Non-Small Cell Lung Cancer

Lung cancer is still the leading cause of cancer death. Non-small cell lung cancer accounts for approximately 85% of all lung cancer cases. Changes in medical practice related to cancer screening and/or treatment resulted in a decline in lung cancer incidence and mortality, as reported [1]. Even though the 5-year survival rate for NSCLC is an average of 26%, compared to 7% for small cell lung cancer (cancer.net). Progress has been made to discover lncRNA biomarkers in lung cancer, such as MALAT-1, HOTAIR, and CCAT2 [23]. Gao et al., in their study, suggested FLAVCR1-AS1 as a novel diagnostic biomarker and therapeutic target [36]. The authors explored the regulatory mechanism describing the association of human NSCLC with the FLVCR1-AS1/miR-573/E2F3 axis as an important signaling pathway in tumorigenesis and progression of the disease. FLVCR1-AS1 is upregulated in the cancer tissue and cell lines, and acting as ceRNA enhances E2F3 expression through the competition of the suppressive role of miR-57 [36]. Silencing FLVCR1-AS1 repressed the proliferation, migration, and invasion of lung cancer cells, as also reported in another study where it has been revealed that FLVCR1-AS1 positively regulates the activation of the Wnt/β-catenin pathway [36,37]. The expression level of downstream targets, including proteins such as CTNNB1, SOX4, CCND1, CCND2, c-MYC, and β-catenin, was decreased. Thus, FLVCR1-AS1 could play a crucial role in the formation and development of lung cancer [37].

### 3.7. Ovarian Cancer

Ovarian cancer is fatal cancer in women. The prognosis of OC remains dismal as the survival rate is about 5 years in 30% of patients. An initial diagnosis of OC is required for more successful treatment [1]. A total of 85% of ovarian cases are represented by serious ovarian cancer. An investigation into the molecular mechanism suggested that the high expression of FLVCR1-AS1 induces OC by regulating the miR-513/YAP1 signaling pathway FLVCR1-AS1 expression was upregulated in OSC tissue, serums, and cell lines resulting in EMT, migration, progression, invasion, and inhibition of apoptosis [38].

### 3.8. Breast Cancer

Breast cancer is one of the most common malignancies in women [1]. Despite significant advances in screening and chemotherapy techniques, the survival rate is still less than 5 years after diagnosis. The incidence rate of breast cancer is rapidly growing. So, the diagnosis of the source of the disease can help in the treatment and prognosis of the disease [69,70]. The role of FLVCR1-AS1 in breast cancer has been reported by Pan et al. (2020). They showed that FLVCR1-AS1 was significantly upregulated in BC cells. The expression of FLVCR1-AS1 was found to be positively correlated with tumor growth, size, and volume in vivo, which supported that FLVCR1-AS1 played an oncogenic role in BC. Furthermore, FLVCR1-AS1 promotes proliferation and migration and activates the Wnt/β-catenin pathway through the miR-381-3p/CTNNB1 axis in breast cancer. Silencing FLVCR1-AS1 inhibited cell proliferation and inhibition of miR-381-3p, which was identified as a potential target gene of FLVCR1-AS1, reversed the tumor-restraining impacts of FLVCR1-AS1 depletion on BC progression [41].

### 3.9. Osteosarcoma

Osteosarcoma is the most common bone cancer in children and young people. Due to the high mortality of this cancer, specific biomarkers associated with OS tumorigenesis, progression, and distant metastasis, including from the lncRNA category, still need to be identified to help the development and implementation of more effective treatments. A great number of lncRNAs implicated in many tumorigenesis signaling pathways are dysregulated in osteosarcoma [71].

A study conducted by Jiang et al. has indicated that LncRNA FLVCR1-AS1 may promote proliferation, migration, and invasion of osteosarcoma cells by regulating the Wnt/β-catenin pathway [43].

### 3.10. Pancreatic Cancer

Pancreatic cancer is one of the most lethal cancer due to late diagnosis, grave prognosis, and high metastasis incidence. A 5-years survival rate is still less than 10% [1]. Moreover, the difficulty in radical surgery and the unmet need for new treatment effective against chemotherapeutic resistance require investigation of underlying mechanisms of PC tumorigenesis to uncover a new target [1]. As we just described for numerous cancers, including lung, breast, and ovarian cancer, FLVCR1-AS1 is a tumor regulator in cell proliferation, migration, and invasion [36,37,38,39]. Lin et al. described positive feedback between FLVR1-AS1 and Kruppel-like factor 10 (KLF10), known as a tumor suppressor by regulating the transcription level of target genes through the PTEN/AKT pathway in multiple cancers, including PC [41,72,73]. They found that FLVCR1-AS1 was significantly suppressed in PC tumor tissues and was associated with poor prognosis. Furthermore, functional studies demonstrated that FLVCR1-AS1 could act as a tumor suppressor by inhibiting PC cells.

Proliferation and migration assays both in vitro and in vivo revealed that FLVCR1-AS1 was directly transcribed by KLF10 and inversely upregulated KLF10 expression by sponging miR-513c-5p and miR-514b-5p and functioning as a ceRNA for KLF10 in PC [41].

## 4. Regulating Mechanisms of FLVCR1-AS1

The key roles in gene regulation of lncRNAs affecting cellular homeostasis leading to tumorigenesis, cancer progression, and distant metastasis have been well established in many studies. Increasing evidence has demonstrated that lncRNAs participate in various cancer development and progression, very often acting as sponges of miRNAs to regulate their target gene expression (Figure 5) [28,29,30,31,32,33,34,35,36,37,38,39,40,41].

It was observed that the level of β-catenin protein that is commonly found in eukaryotic cells was altered in many types of cancers. Additionally, it has been proven that Wnt/β-catenin is one of the main pathways in lung cancer progression [72]. Lin et al. (2019) analyzed the expression of target genes in the Wnt/β-catenin pathway, including *CTNNB1*, *SOX4*, *CCND1*, *CCND2*, and *c-MYC*. This study’s findings showed that target gene expression levels were downregulated after FLVCR1-AS1 silencing and had stopped cell growth and proliferation. Thus, FLVCR1-AS1 can induce the proliferation, migration, and invasion of lung cancer cells by activating the Wnt/β-catenin signaling pathway (Table 3) [37].

Furthermore, Zhang et al. (2018) revealed that highly expressed FLVCR1-AS1 in HCC could decrease miR-513c expression while increasing MET expression, thus suggesting that FLVCR1-AS1 promotes tumorigenesis through modulating MET [30]. In the study of gastric cancer, it was suggested that FLVCR1-AS1 regulates c-Myc expression through miRNA sponging [31]. The c-Myc is a significant miR-155 target gene and is a known proto-oncogene since its high expression is associated with cell proliferation, cancer development, and poor prognosis. Thus, an FLVCR1-AS1-miR-155-c-Myc signaling axis has been identified in gastric cancer. Moreover, the silencing effect of FLVCR1-AS1 on high expression of p21 stops cell growth and improves cell apoptosis [31]. The E2 factor (E2F) family of transcription factors are downstream effectors of the cyclin-dependent kinase (CDK)–RB–E2F. It has been described as an oncogene in many cancers [75]. E2F- transcription factor 2 in glioma and E2F-transcription factor 3 in lung cancer are related to the high proliferation and invasion by the role of the molecular sponge of FLVCR1-AS1 for miR-4731-5p and miR-57, respectively [34,36]. In ovarian cancer, the high expression of YAP1 observed confirmed that miR-513 directly binds to YAP1 and that miR-513 has an inverse relationship with FLVCR1-AS1 [38].

## 5. The Effect of FBXL19-AS1 on Several Cancers

### 5.1. Hepatocellular Carcinoma

An experimental study conducted by Hao et al. illustrated that the LncRNA FBXL19-AS1/miR-342-3p pathway plays a crucial function in the proliferation, migration, and invasion of Huh7 cells treated by flavonoids of Sophorae Fructus [42]. Furthermore, another study revealed a positive correlation between FBXL19-AS1 TNM stage and poor prognosis of HCC patients, where FBXL19-AS1 acts as a regulation agent, controlling HCC-associated pathways, including cell cycle, microRNAs in cancer, viral carcinogenesis via ceRNA network [43]. Consequently, this lncRNA may be considered a diagnostic and prognostic biomarker [44]. Furthermore, a study demonstrated that lncRNA FBXL19-AS1 has a diagnostic biomarker potential and is considered an appropriate candidate as a prognosis biomarker in HCC [45].

### 5.2. Gastric Cancer

It has been confirmed that lncRNA FBXL19-AS1 could be upregulated in gastric cancer [46,47]. Wang et al. showed that lncRNA FBXL19-AS1 could play a key role as a competing endogenous RNA to control ZEB1 expression by sponging miR-431 in gastric cancer [46]. Hence, FBXL19-AS1 may serve as a novel potential target for cancer therapy in gastric cancer. A recent study found that lncRNA FBXL19-AS1 has high expression in tumor tissues and the progress of gastric cancer via modulating miR-876-5p/HMGB4 axis [47].

### 5.3. Colorectal Cancer

In the study of Shen et al., 2439 lncRNAs and 1654 mRNAs were differentially expressed in metastatic CRC relative to primary CRC. Among these dysregulated lncRNAs, FBXL19-AS1 was the most significantly upregulated lncRNA in metastatic tumors. Furthermore, overexpression of FBXL19-AS1 was significantly associated with increased cell proliferation and invasion in colorectal tissue, acting as a molecular sponge in negatively modulating miR-203 [48]. Consequently, the expression of this lncRNA can be associated with colorectal cancer progression.

### 5.4. Glioma

Liu et al. revealed an interesting mechanism of IGF2BP2/FBXL19-AS1/ZNF765 axis on blood-tumor-barrier (BTB) permeability through tight junction-associated proteins [49]. Their study found that insulin-like grown factor 2 mRNA-binding protein 2 (RBP-IGF2BP2) and FBXL19-AS1 were overexpressed in glioma microvessel and glioma endothelial cells (GECs). In contrast, ZNF765 was lowly expressed in the comparison of ECs. When the transcription factor ZNF765 is overexpressed, it decreases the BTB permeability by inhibiting the promoter activities of tight junction-related proteins, ZO-1, occludin, and claudin-5. In this study, the authors demonstrated further the inhibition by ZNF765 on the transcriptional activity of IGF2BP2, which is shown to stabilize the expression of lnc-FBXL19-AS1 as well. In the meantime, overexpression of FBXL19-AS1 reduced the half-life of transcription factor ZNF765 mRNA via the SMD pathway and further accelerated its degradation. Consequently, FBXL19-AS1 may modulate the blood-tumor barrier permeability by negatively controlling ZNF765 via STAU1-induced mRNA decay [49].

### 5.5. Lung Cancer

A recent study showed that the high expression of FBXL19-AS1 drives cell proliferation and growth by regulating epithelial–mesenchymal transition (EMT) in NSCLC [50]. FBXL19-AS1 was also shown to exert a tumorigenic role in leading the progression and angiogenesis by sponging miR-431-5p to regulate RAF1 expression in lung cancer [51]. Furthermore, cell cycle analysis revealed that FBXL19-AS1 knockdown could stop the growth of tumors in the G0/G1 phase in vitro and in vivo. FBXL19-AS1/miR-203a-3p axis was found to associate with baculoviral IAP repeat-containing protein 5.1-A-like (survivin), distal-less homeobox 5, E2F transcription factor 1, and zinc finger E-box binding homeobox 2 to regulate metastasis in LUAD cells [52].

### 5.6. Cervical Cancer

Cervical cancer is the most preventable cancer. Unfortunately, it is persistently the second leading cause of cancer death in women aged 20 to 39. Diagnoses among young women are increasing the incidence of advanced disease and cervical adenocarcinoma, for which cytology is less effective at prevention and early detection than squamous cell carcinoma [1]. A research study investigated the role of lncRNA FBXL19-AS1 on cervical cancer proliferation, migration, invasion, apoptosis, and EMT [53]. This study demonstrated that LncRNA FBXL19-AS1 induces metastasis and proliferation of CC by acting as a ceRNA and inhibits the expression of miR-193a-5p, which negatively regulate COL1A1 expression [53]. Wan et al. indicated that FBXL19-AS1 might play an essential role in cervical cancer formation by affecting cell growth, migration, and invasion via miR-193a-5p/PIN1 signaling [54].

### 5.7. Breast Cancer

Recent research evaluated the functional and mechanistic roles of FBXL19-AS1 and suggested that high levels of FBXL19-AS1 expression promote migration and invasion of breast cell lines and can affect breast cancer by regulating miR-718 expression. FBXL19-AS1 might act as a diagnostic biomarker for breast cancer [25,55]. Moreover, it was demonstrated that FOXM1 upregulation inhibits tumor growth by silencing FBXL19-AS1 and that the FBXL19-AS1/miR-8765p/FOXM1 axis might regulate breast cancer cell proliferation [56]. Based on the studies’ results so far, it is possible that FBXL19-AS1 may play an important role in the tumor genesis and progression of breast cancer [55,56,57].

### 5.8. Osteosarcoma

A study revealed that there is an inverse relationship between FBXL19-AS1 and miR-346, where FBXL19-AS1 acts as an oncogenic marker, inducing proliferation and invasion in OS [58].

### 5.9. Nasopharyngeal Carcinoma

Nasopharyngeal carcinoma is a harmful and aggressive malignant tumor beginning in the nasopharyngeal epithelium [59]. Dong et al. demonstrated a relation between FBXL19-AS1 and miR-431 and PBOV1 [59]. It was identified that FBXL19-AS1 could induce the progress of nasopharyngeal carcinoma by serving as a competing endogenous RNA to sponge miR-431 and upregulate PBOV1. This study suggested that FBXL19-AS1 could be a potential therapeutic target for NPC patients. However, more studies are needed to better understand the mechanisms of action FBXL19-AS1 in NPC [59].

### 5.10. Acute Myeloid Leukemia

The dysregulated expression of FBXL19-AS1 has been considered related to acute myeloid leukemia. Sheng et al. demonstrated that the increased expression of FBXL19-AS1 in acute myeloid leukemia is related to clinicopathological factors, including cytogenetics (=0.021) and French-American-British classification (= 0.011) [60]. In addition, patients with upregulated FBXL19-AS1 expression had relatively short overall survival and disease-free survival. It seems that FBXL19-AS1 can be used as a potential biomarker in the diagnosis of AML [60]. However, this hypothesis needs further study and investigation in the future.

## 6. Molecular Mechanisms of FBXL19-AS1

Many studies have revealed that FBXL19-AS1 can regulate the expression of specific miRNAs that play an oncogenic role in various cancers. FBXL19-AS1 causes cancers by acting in different signaling pathways (Figure 6; Table 4) [42,43,44,45,46,47,48,49,50,51,52,53,54,55,56,57,58,59,60].

The EMT signaling pathway is important in cell proliferation and migration. EMT signaling pathway could be responsible for high cell proliferation and cancer development. FBXL19-AS1 can induce the EMT signaling pathway to elevate breast cancer progression through changes in the expression of EMT-related genes, such as N-cadherin and E-cadherin [57]. FBXL19-AS1 can affect cell proliferation and invasion by sponging different miRNAs [42,46,47,48,49,51,52,53,54,55,56,59]. A study demonstrated that lncRNA FBXL19-AS1 was highly expressed in breast cancer and promoted the proliferative and invasive potentials of breast cancer cells by functioning as a molecular sponge of miR-718. So, it was found that miR-718 was significantly decreased and inversely associated with FBXL19-AS1 expression in breast cancer tissues. Furthermore, they showed that FBXL19-AS1 inhibition significantly induced the expression of epithelial marker E-cadherin and decreased mesenchymal marker N-cadherin in mRNA and protein levels [55]. FBXL19-AS1 as an oncogene can also develop breast cancer through other pathways since the upregulation of FBXL19-AS1 accelerates the expression of Forkhead box M1 (FOXM1) by sponging of miR-876-5p [56]. Pan et al. (2018) suggested a critical role of FBXL19-AS1/miR-346 in osteosarcoma cell proliferation, migration, and invasion, and so inhibition of miR-346 can increase osteosarcoma [58]. Shen et al., by microarray analysis, report that 2439 lncRNAs and 1654 mRNAs were differentially expressed in metastatic CRC relative to primary CRC. Among these dysregulated lncRNAs, FBXL19-AS1 was the most significantly upregulated lncRNA in metastatic tumors [60]. Bioinformatics analysis predicted that miR-203 was potentially targeted by FBXL19-AS1, confirmed by a dual-luciferase reporter assay [52]. So, they suggested that FBXL19-AS1 may be a potential oncogenic lncRNA in CRC and FBXL19-AS1/miR-203 regulatory pathway involved in CRC tumorigenesis. Another study proved that FBXL19-AS1/miR-431-5p/RAF1 axis functions as an oncogenic pathway and plays an effective role in lung cancer. In more detail, upregulation of FBXL19-AS1 could decrease miR-431 and increase RAF1 kinase activity, which in turn increased polarity and tumor growth in colorectal cancer [52].

## 7. Conclusions

The remarkable advance in the field of cancer demonstrates the critical role of lncRNAs for their act as diagnostic or prognostic biomarkers in different types of cancer. Many lncRNAs that have demonstrated key target agents offer a promising approach for cancer treatment and diagnosing diverse cancer diseases in preclinical and clinical conditions. On the other hand, LncRNAs stand out for their great specificity in tumor tissues and cells, which enables them to be precise and specific biomarkers. Additionally, non-invasive extraction of abnormally expressed lncRNAs has the potential to be both more beneficial and cost-effective. Given that lncRNA are expressed at low levels and are less hazardous than protein-based anti-tumor medications, only a minimal concentration of inhibitors is required to have an impact. Additionally, the discovery of lncRNA biomarkers is given additional prospects by bioinformatics and computational techniques. Several studies have verified that lncRNAs FLVCR1-AS1 and FBXL19-AS1 can act as an oncogene in the incidence and progress of different types of human cancers.

FLVCR1-AS1 upregulated in patient tissues with multiple different of human cancer, including cholangiocarcinoma, gastric cancer, glioma and glioblastoma, hepatocellular carcinoma, lung cancer, ovarian cancer, breast cancer, and colorectal cancer, is of great importance due to its regulatory role on several signaling pathways related to cancer initiation and development. So, the upregulation of FLVCR1-AS1 is involved in various aspects of tumorigenesis, including cell growth, proliferation, migration, invasion, and EMT.

Furthermore, the results of various studies have shown the controlling potential of FBXL19-AS1 as a putative lncRNA candidate in several human cancers, including breast cancer, colorectal, lung, and osteosarcoma. Furthermore, dysregulation of FBXL19-AS1 is mediated in various features of tumor formation, including cell growth, proliferation, migration, invasion, and EMT.

Among the lncRNAs known so far, PCA3 is the first and only lncRNA to receive Food and Drug Administration (FDA)-approval as a cancer biomarker test to date. PCA3 is a prostate-specific marker often overexpressed in prostate cancer and can be detected easily through non-invasive urine collection. Uncovering new and sensitive biomarkers and therapeutic targets will be made possible by a thorough understanding of lncRNA’s expression, structure, and processes. Given the promising results regarding the role of FLVCR1-AS1 and FBXL19-AS1 in cancer, we can hope for their cancer biomarker role in the future. However, additional research, including in vivo tests and clinical studies, is required to more fully substantiate the biofunctions of FLVCR1-AS1 and FBXL19-AS1 in cancer.

## Figures and Tables

**Figure 1 ncrna-09-00001-f001:**
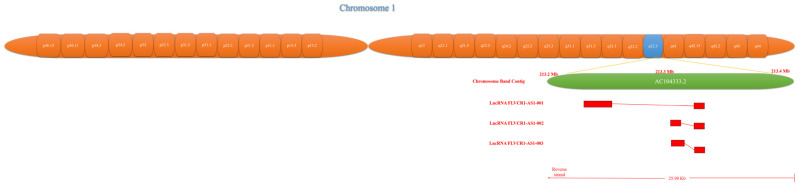
Genomic location of FLVCR1-AS1.

**Figure 2 ncrna-09-00001-f002:**
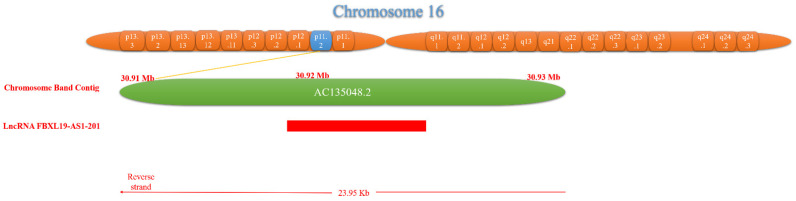
Genomic location of FBXL19-AS1.

**Figure 3 ncrna-09-00001-f003:**
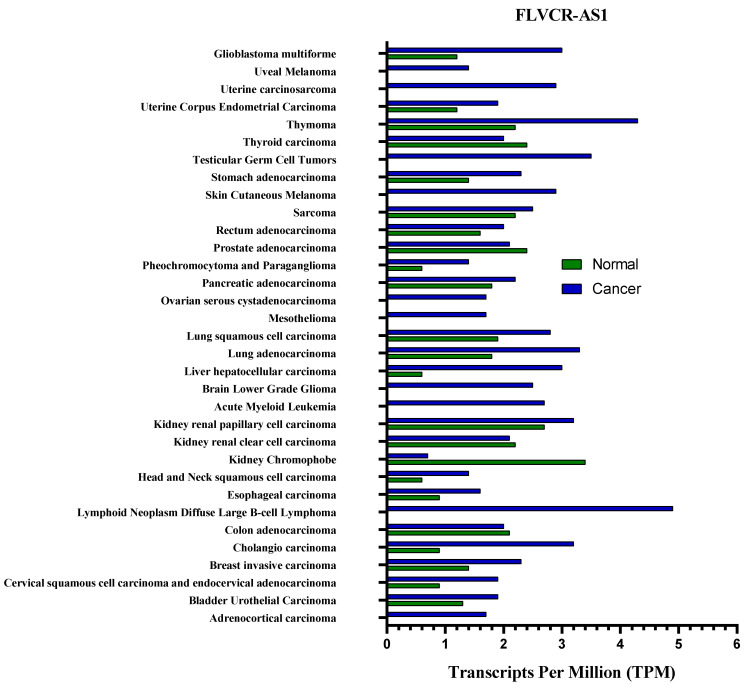
The gene expression of FLVCR1-AS1 profile across all tumor samples and paired normal tissues.

**Figure 4 ncrna-09-00001-f004:**
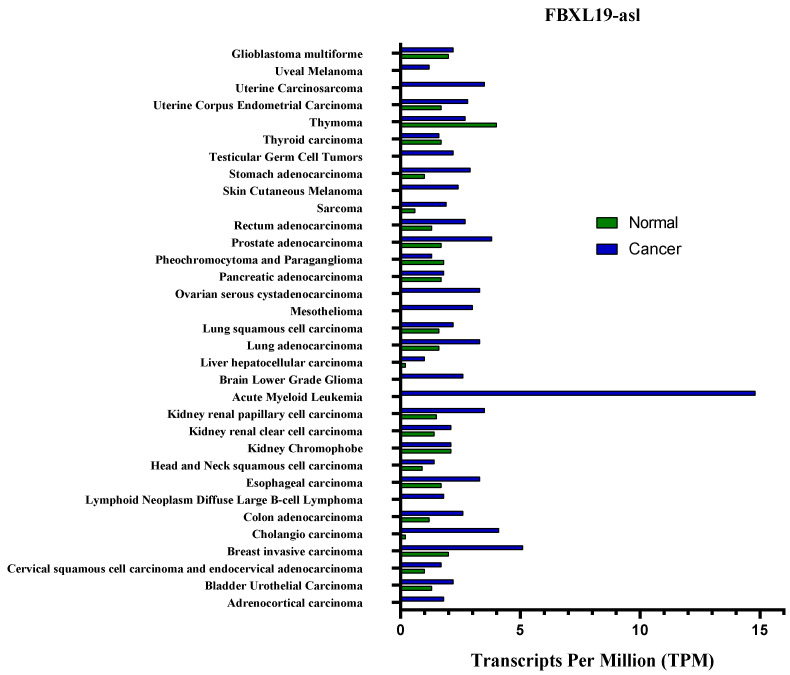
The gene expression of FBXL19-AS1 profile across all tumor samples and paired normal tissues.

**Figure 5 ncrna-09-00001-f005:**
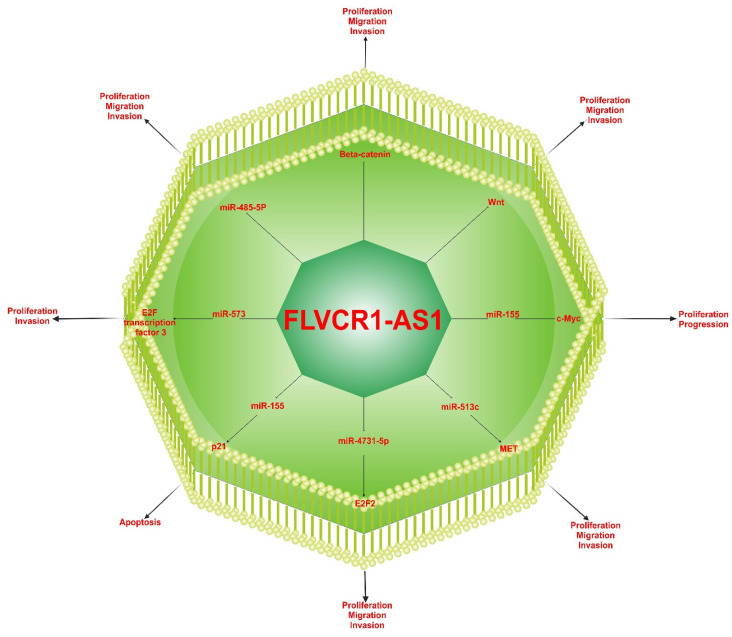
Underling molecular mechanisms of FLVCR1-AS1.

**Figure 6 ncrna-09-00001-f006:**
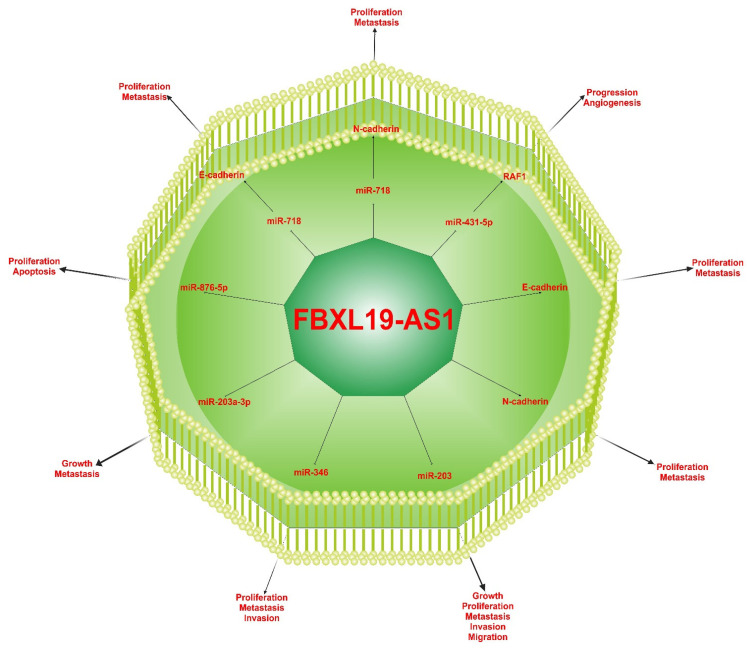
Underling molecular mechanisms of FBXL19-AS1.

**Table 1 ncrna-09-00001-t001:** Analysis of the subcellular localization of FLVCR1-AS1 (from lnclocator database). Database website: http://www.csbio.sjtu.edu.cn/bioinf/lncLocator2/ (accessed on 26 January 2021).

Subcellular Locations	Score
Cytoplasm	0.0682472744361
Nucleus	0.0229142140504
Ribosome	0.0409735673257
Cytosol	0.828903123455
Exosome	0.0389618207329

**Table 2 ncrna-09-00001-t002:** Analysis of the subcellular localization of FBXL19-AS1 (from lnclocator database). Database website: http://www.csbio.sjtu.edu.cn/bioinf/lncLocator2/ (accessed on 26 January 2021).

Subcellular Locations	Score
Cytoplasm	0.323765378524
Nucleus	0.170541522227
Ribosome	0.275374297237
Cytosol	0.075880246739
Exosome	0.154438555273

**Table 3 ncrna-09-00001-t003:** Functional characterizations of FLVCR1-AS1 in multiple human cancers. ↑ in the table indicates upregulation.

Cancer Type	Number of Case	Assessed Cell Lines	Interacting Genes and Proteins	Novel Therapeutic/Diagnostic	Expression	Function	Years Reference
Gastric cancer	A total of 30 Gastric cancer tissues and their adjacent normal tissues	The human gastric epithelial cell line GES-1 and three human gastric cancer cell lines (AGS, MNK-45, and MGC-803)	miR-155, c-Myc	A novel therapeutic target for treatment of patients with GC	↑	Promote cell proliferation and invasion	2019 [31]
Hepatocellular carcinoma	A total of 60 HCC and matched normal tissues	The normal liver cell line LO2 and the HCC cell lines(Hep3B, HepG2, Huh7, and PLC/PRF-5)	miR-513c	A new target in HCC prevention and treatment	↑	Promote cell proliferation, migration, and invasion	2018 [30]
Lung cancer	A total of 62 NSCLC and adjacent normal lung tissues	Five NSCLC cell lines (H358, A549, H520, H1299, and SKMES1), human bronchial epithelial cell line (BEAS-2B), and human embryonic kidney (HEK) 293T cell line	miR-573, E2F transcription factor 3	A novel diagnostic biomarker and therapeutic target for NSCLC	↑	Promote cell proliferation and invasion	2018 [36]
Lung cancer	A total of 29 lung cancer tissues and their adjacent normal tissues	Human bronchial epithelial cell line (16HBE) and human lung cancer cell lines (SPCA1, A549, and H1299)	Wnt/β-catenin signaling pathway	A novel targets for the treatment of lung cancer	↑	Promote cell proliferation, migration, and invasion	2019 [37]
Glioma	A total of 51 glioma tissues and adjacent normal tissues	Glioma cell lines and human astrocyte cell line (NHA)	miR-4731-5p/E2F2	A potential target for glioma tumor therapy	↑	Promote cell proliferation, migration, and invasion	2020 [34]
Glioma	A total of 50 glioblastoma tissues and adjacent normal tissues	The human GBM cell lines U251, T98G, LN229 and SHG44 and normal human astrocyte (NHA) cells	miR-30b-3p	A novel therapeutic target and diagnostic biomarker for glioblastoma	↑	Promote cell proliferation and invasion	2020 [35]
Breast cancer	-	Human normal breast epithelial cell (MCF-10A) and BC cells (MDA-MB-231, T47D, BT-474, SKBR3, MCF7)	Wnt/β-catenin pathway, miR-381-3p, CTNNB1	A promising target for breast cancer therapy	↑	Promote cell proliferation, migration, invasion	2020 [39]
Ovarian cancer	A total of 50 ovarian serous cancer tissues and adjacent normal tissues	SKOV3 and OVCAR3 cells	miR-513, YAP1	A potential therapeutic target for human ovarian cancer	↑	Promote cell progression, migration, invasion, and EMT process	2019 [38]
Cholangiocarcinoma	A total of 22 cholangiocarcinoma and adjacent normal tissues	Human cell lines, including the noncancerous cholangiocyte cell line HIBEC and the CCA cell lines RBE, CCLP1, HuCCT1, and HCCC-9810	miR-485-5p	A novel therapeutic target and a potential diagnostic marker for cholangiocarcinoma	↑	Promote cell proliferation, migration, and invasion	2019 [28]
Colorectal cancer	A total of 26 pairs of colorectal cancer tissues and adjacent non-tumor tissues	Four human CRC cell lines, namely Caco-2, SW480, LoVo, and SW1116, the NCM460 normal colonic epithelial cell line and 293T cells	miR-381, RAP2A	-	↑	Promote cell viability, apoptosis, migration, and invasion	2020 [32]
Colorectal cancer	-	Human CRC cell lines, namely Caco-2, SW480	miR-493-3p	-	↑	Promote cell proliferation, invasion, and migration	2020 [33]
Osteosarcoma cells	A total of 48 osteosarcoma tissues and adjacent non-tumor tissues	Normal osteoblasts (hFOB1.19) and U2OS and MG63 osteosarcoma cell lines	Wnt/β-catenin pathway, CTNNB1, SOX4, CCND1, CCND2, MYC and nucleus β-catenin	-	↑	Promote cell proliferation, invasion, and migration	2020 [40]
pancreatic cancer	A total of 77 samples of human pancreatic cancer tissues and corresponding normal tissues	Human PC cell lines (Bxpc-3, CFPAC-1, MIA PaCa-2,PANC-1, and PATU-8988) and human normal pancreatic ductal epithelial cells (HPNE)	KLF10 and PTEN/AKT pathway	A noveltherapeutic strategy for PC treatment	↑	Inhibit proliferation, cell cycle, and migration	2021 [41]
Cervical cancer	-	Cervical cancer cell	miR-381, MAGT1	-	↑	Promote cell growth	2022 [74]

**Table 4 ncrna-09-00001-t004:** Functional characterizations of FBXL19-AS1 in multiple human cancers. ↑ in the table indicates upregulation.

Cancer Type	Number of Case	Assessed Cell Lines	Interacting Genes and Proteins	Novel Therapeutic/Diagnostic	Expression	Function	Years Reference
Gastric cancer	-	-	miR-876-5p, HMGB3	-	↑	Promote cell development	2020 [47]
Gastric cancer	-	-	miR-431, ZEB1	-	↑	-	2020 [76]
Breast cancer	A total of 49 breast cancer tissues and adjacent normal tissues	Human BC cell lines (MDA-MB-231, ZR-75-1, MCF-7, BT-549, MDA-MB-468, and T47D) and the normal mammary fibroblast cell line (Hs578Bst)	miR-718	A potential therapeutic target for breast cancer treatment	↑	Promote cell proliferation and invasion	2019 [55]
Breast cancer	-	Non-carcinogenic epithelial cells (MCF-10A) and breast cancer cells (MCF-7, BT-549, MDA-MB-231, and SKBR3)	miR-876-5p, Forkhead box M1 (FOXM1)	A therapeutic approach for treating breast cancer	↑	Promote cell proliferation and apoptosis	2019 [56]
Breast cancer	-	Normal cell line HS578Bst, breast cancer cell lines (SK-BR3, BT474, MCF-7, and MDA-MB-231)	lin-28 homolog A (LIN28A), WD repeat domain 66 (WDR66)	A new biological marker in breast cancer	↑	Promote cell migration, invasion, and EMT	2019 [57]
Colorectal cancer	A total of 50 human colorectal cancer tissues and their adjacent non-tumor tissues	CRC cell lines (LoVo,HT29,HCT116, and SW620) and normal colon epithelial FHC cel	miR-203	A new insight for understanding CRC development	↑	Promote cell proliferation, migration, and invasion	2017 [48]
Lung cancer	A total of 84 lung cancer tissues and adjacent non-tumor tissues	Lung cancer cell lines (A549, H1975, SPC-A-1, H125, and H1299) and normal human lung cells (MRC-5)	miR-431-5p, RAF1	A new insight into the therapeutic strategies of lung cancer	↑	Promote cell proliferation, migration, invasion, and angiogenesis	2019 [51]
Non-small cell lung cancer	A total of 52 lung cancer tissues and adjacent non-tumor tissues	Five kinds of NSCLC cell lines (A549, H1299, H520, SPCA1, and H358) and normal lung epithelial cells (16HBE)	E-cadherin, N-cadherin, vimentin	-	↑	Promote cell proliferation and metastasis	2019 [50]
Lung adenocarcinoma	The lung adenocarcinoma and matched normal adjacent tissue samples	Two human lung adenocarcinoma cells, NCI-H1975 and SPC	miR-203a-3p	A potential prognostic marker and a therapeutic target for patients with lung adenocarcinoma	↑	Promote tumor growth and metastasis	2020 [52]
Osteosarcoma	-	Human osteosarcomacell lines MG63, U2OS, SAOS2, HOS, 143B, and the normal osteoblast cell line hFOB1.19	miR-346	A novel therapeutic target for osteosarcoma	↑	Promote cell proliferation, migration, and invasion	2018 [58]
Cervical cancer	A total of 100 cervical cancer tissues and adjacent non-tumor tissues	Human normal cervical cell line (Ect1/E6E7) and cervical cancer cell lines (C-4-I, SiHa, C-33A and HeLa)	miR-193a-5p, PIN1	A new direction for treating patients with cervical cancer	↑	Promote cell growth, migration, and invasion	2020 [54]
Cervical cancer	A total of 46 cervical cancer tissues and adjacent non-tumor tissues	Human healthy cervical cells (HUCEC) and human cervical cancer cells (HeLa, Caski, C-33 A, AV3)	COL1A1, miR-193a-5p	-	↑	Promote cell proliferation and metastasis	2021 [53]
Hepatocellular carcinoma	A total of 57 hepatocellular carcinoma tissues and adjacent non-tumor tissues, and whole blood samples of 92 patients with hepatocellular carcinoma	-	-	A potential biomarker for HCC diagnosis and prognosis	↑	Promote cell occurrence and development	2020 [44]
Hepatocellular carcinoma	-	-	miR-342-3p	-	↑	Promote cell proliferation, migration, and invasion	2020 [42]
Hepatocellular carcinoma	A total of 60 hepatocellular carcinoma tissues and adjacent non-tumor tissues	Normal liver cell line and HCC cell lines	KLF2	-	↑	Promote tumor progression	2021 [45]
Hepatocellular carcinoma	A total of hepatocellular carcinoma tissues and adjacent non-tumor tissues	Normal liver cell line and HCC cell lines	miR-541-5p	-	↑	Promote cell proliferation, invasion, and migration	2021 [77]
Glioma	-	The human brain microvascular endothelial cell line hcMEC/D3 (ECs)	ZNF765, STAU1	A new potential therapeutic strategy for glioma	↑	Promote cell apoptosis	2020 [49]
Nasopharyngeal carcinoma	A total of 30 nasopharyngeal carcinoma tissues and adjacent non-tumor tissues	Human NPC cell lines (C666-1, SUNE1, 5–8F and 6–10B) and nasopharyngeal epithelial cells (NP69)	miR-431, PBOV1	A novel therapeutic target for nasopharyngeal carcinoma	↑	Promote tumor progression	2021 [59]
Acute myeloid leukemia	Serum samples of 137 acute myeloid leukemia patients and 43 healthy controls	-	-	A novel prognostic and diagnostic biomarker for acute myeloid leukemia patients	↑	-	2021 [60]
Pancreatic cancer	A total of 73 pancreatic cancer tissues and adjacent non-tumor tissues	Normal pancreatic epithelial cells (hTERT-HPNE) and pancreatic cancer cell lines (Capan-1, SW1990, PaTu8988)	miR-339-3p	-	↑	Promote cell proliferation, migration, and invasion	2021 [78]

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
