# Peer review of "FLVCR1-AS1 and FBXL19-AS1: Two Putative lncRNA Candidates in Multiple Human Cancers"

_ncrna, 2022, doi:10.3390/ncrna9010001_

Round 1
Reviewer 1 Report
In the current review, the author summarized the roles of two lncRNAs, FLVCR1-AS1 and FBXL19-AS1, in multiple cancer types. The content is solid with many convincing references. I still have some suggestions hope can help the author to improve the manuscript.
1. The use of abbreviations needs to be revised in the whole manuscript.
2. Gene and protein nomenclature needs to be revised. Gene names may need to be italicized.
3. Grammar and typo issues are commonly found across the manuscript and need to be carefully revised. (example: line 80, missing space between “FLVCR1-AS1” and “gene”)
4. In the introduction part, the author introduced the mechanisms of lncRNAs by interacting with protein, DNA, and RNA. The author gave explanations of how lncRNAs interact with protein (enzymes), and RNA (miRNAs), but did not clearly explain how lncRNAs interact with DNA. May need to revise the statements on line 67-69.
5. A bit of background introduction about why these two lncRNAs were picked for review would be great.
6. The links for lncRNA info at ensemble.com are invalid. Need double check. The author also mentioned variant splice, but did not explains why this matter for lncRNA functions.
7. The author mentioned about the subcellular distribution of the two lncRNAs (lin 88 and 101) in the content but did not mention it in the tables. May need to add the information and revise tables 1 and 2.
8. I would suggest re-organizing Table 1 and 2. Use symbols (such as replace upregulation with symbol ↑) to make the table less busy. For the “Related genes and proteins” table, does it mean “interacting genes and proteins”?
9. The explanation of lncRNA function in Table 1 and 2 is oversimplified. Better to specify whether the lncRNA is promoting or inhibiting proliferation. By reading the content below, it seemed that the same lncRNA may play opposite roles in controlling cell proliferation based on the cancer type. So, it is very important to point out clearly how the lncRNA act in the specific cancer type in the summary table.
10. In Figure 3 and 4, I would suggest using the organ/tissue name to replace the cancer abbreviations. The purpose of these two figures is to compare the expression of lncRNAs in tissues between normal and cancerous conditions. The use of the cancer name on the Y-axis would suggest the tissue is already cancerous.
11. In line 169, the author also stated that “the lncRNA FLVCR1-AS1 acts as an oncogene”, which is contradictory to the above (line 89).
12. In Figure 5 and 6, it is better to specify whether the lncRNA is promoting or inhibiting these pathways.
13. Line 305 – 307. Table 3 seemed irrelevant to the statement about the FLVCR1-AS1 function.
14. Similarly, line 422-423, table 4 is not relevant to the FBXL19-AS1 function.
Author Response
Response to reviewers’ comments
We wish to express our appreciation to the Reviewers for their insightful comments, which have helped us significantly to improve our manuscript. According to the suggestions, we have thoroughly revised our manuscript and its final version is enclosed. Point-by-point responses to the comments are listed below.
Reviewer 1:
- The use of abbreviations needs to be revised in the whole manuscript.
Response: We thank the reviewer for allowing us to revise the abbreviations in the whole manuscript.
- Gene and protein nomenclature needs to be revised. Gene names may need to be italicized.
Response: In accordance with the reviewer's comment, we have revised Gene and protein nomenclature in the whole manuscript.
- Grammar and typo issues are commonly found across the manuscript and need to be carefully revised. (example: line 80, missing space between “FLVCR1-AS1” and “gene”)
Response: We are grateful to the reviewer for this point on this issue and in accordance with the reviewer's comment, we have revised Grammar and typo.
- In the introduction part, the author introduced the mechanisms of lncRNAs by interacting with protein, DNA, and RNA. The author gave explanations of how lncRNAs interact with protein (enzymes), and RNA (miRNAs), but did not clearly explain how lncRNAs interact with DNA. May need to revise the statements on line 67-69.
Response: We appreciated this comment and according to it we gave further details about the DNA-LncRNA interaction and accordingly to the referee’s comments we have revised the statements on line 67-69.
- A bit of background introduction about why these two lncRNAs were picked for review would be great.
Response: In accordance with the reviewer's comment, we have revised the background introduction.
- The links for lncRNA info at ensemble.com are invalid. Need double check. The author also mentioned variant splice, but did not explains why this matter for lncRNA functions.
Response: We thank the referee and apologize for the inconvenience. We double-checked the links for lncRNA info at ensemble.com from different resources and in our systems works properly.. We also added a few senteces about the variant splice role in LncRNAs. The author mentioned about the subcellular distribution of the two lncRNAs (lin 88 and 101) in the content but did not mention it in the tables. May need to add the information and revise tables 1 and 2.
Response: We appreciate this comment because it allowed us to be more compelling to the reader and in accordance with the reviewer's comment, we have added Table 1 and Table 2 to the manuscript.
- I would suggest re-organizing Table 1 and 2. Use symbols (such as replace upregulation with symbol ↑) to make the table less busy. For the “Related genes and proteins” table, does it mean “interacting genes and proteins”?
Response: We thank the referee for this comment giving us the opportunity to improve the tables. We have then revised and named Table 3 and Table 4. We also replaced “Related genes and proteins” table, with “interacting genes and proteins”
- The explanation of lncRNA function in Table 1 and 2 is oversimplified. Better to specify whether the lncRNA is promoting or inhibiting proliferation. By reading the content below, it seemed that the same lncRNA may play opposite roles in controlling cell proliferation based on the cancer type. So, it is very important to point out clearly how the lncRNA act in the specific cancer type in the summary table.
Response: We appreciate this comment and after revision we strongly believe to meet the referee’s expectation to point out clearly how the lncRNA act in specific cancer type.
- In Figure 3 and 4, I would suggest using the organ/tissue name to replace the cancer abbreviations. The purpose of these two figures is to compare the expression of lncRNAs in tissues between normal and cancerous conditions. The use of the cancer name on the Y-axis would suggest the tissue is already cancerous.
Response: We thank the referee and In accordance we have revised the Figure 3 and 4.
- In line 169, the author also stated that “the lncRNA FLVCR1-AS1 acts as an oncogene”, which is contradictory to the above (line 89).
Response: In accordance with the reviewer's comment, paper have revised.In Figure 5 and 6, it is better to specify whether the lncRNA is promoting or inhibiting these pathways.
Response: In accordance with the reviewer's comment, we have revised the Figure 5 and 6.
- Line 305 – 307. Table 3 seemed irrelevant to the statement about the FLVCR1-AS1 function.
Response: In accordance with the reviewer's comment, we have revised the Table 3.
- Similarly, line 422-423, table 4 is not relevant to the FBXL19-AS1 function.
Response: In accordance with the reviewer's comment, we have revised Table 4.
Reviewer 2 Report
Sheykhhasan et al. provide a comprehensive overview of two lncRNAs: FLVCR1-AS1 and FBXL19-AS1 in various cancers. They describe what is known in the literature about the associations of these lncRNAs with cancer and potential molecular mechanisms involved. I have some minor comments.
11. One of the columns in Table 1 and 2 is entitled “Correlation with patients outcome”. However, in most cases notes in the table indicate a “novel therapeutic/diagnostic marker”. This is not “patient outcome”. The column heading should be rephrased adequately.
22. Vast majority of the studies mentioned in the manuscript report those lncRNAs acting as sponges for several miRNAs. Often evidence for such role is superficial. The authors should critically comment on the potential role of lncRNAs as miRNA sponges in the discussion. In general discussion is a bit of a repetition of results, while it should provide a more critical look at the data and discuss future perspectives.
33. Line 54: why is 1000 bp regarded as the upper length for lncRNA? I do not recall such definition.
44. Line 64: “LncNAs can also act as decoys for transcription factors by attaching miRNAs” – this is wrong. What is the relation between decoy for a TF and sponging miRNA?
55. Line 66: the sentence is grammatically incorrect, lacking sense.
66. Line 70-71: also unclear sentence.
77. Figure 1 – resolution is too low, text cannot be read.
Author Response
Response to reviewers’ comments
We wish to express our appreciation to the Reviewers for their insightful comments, which have helped us significantly to improve our manuscript. According to the suggestions, we have thoroughly revised our manuscript and its final version is enclosed. Point-by-point responses to the comments are listed below.
Reviewer 2:
- One of the columns in Table 1 and 2 is entitled “Correlation with patients outcome”. However, in most cases notes in the table indicate a “novel therapeutic/diagnostic marker”. This is not “patient outcome”. The column heading should be rephrased adequately.
Response: We thank again for allowing us to improve the tables. we have revised the Table 1 and 2 as suggested.
- Vast majority of the studies mentioned in the manuscript report those lncRNAs acting as sponges for several miRNAs. Often evidence for such role is superficial. The authors should critically comment on the potential role of lncRNAs as miRNA sponges in the discussion. In general discussion is a bit of a repetition of results, while it should provide a more critical look at the data and discuss future perspectives.
Response: In accordance with the reviewer's comment, we have revised the discussion.
- Line 54: why is 1000 bp regarded as the upper length for lncRNA? I do not recall such definition.
Response: In accordance with the reviewer's comment, we have revised the length for lncRNA.
- Line 64: “LncNAs can also act as decoys for transcription factors by attaching miRNAs” – this is wrong. What is the relation between decoy for a TF and sponging miRNA?
Response: In accordance with the reviewer's comment, we have revised the Line 64.
- Line 66: the sentence is grammatically incorrect, lacking sense.
Response: In accordance with the reviewer's comment, we have revised the Line 66.
- Line 70-71: also unclear sentence.
Response: In accordance with the reviewer's comment, we have revised the Line 70-71.
- Figure 1 – resolution is too low, text cannot be read.
Response: As requested, we have changed the Figure 1 resolution.